# DNA Methylation: A Potential Mediator of the Memory Regulatory Effects of taVNS

**DOI:** 10.3390/cells14171327

**Published:** 2025-08-27

**Authors:** Pak On Patrick Yee, Ka Chun Tsui, Man Lung Fung, Boon Chin Heng, Ersoy Kocabicak, Ali Jahanshahi, Yasin Temel, Arjan Blokland, Luca Aquili, Allan Kalueff, Kah Hui Wong, Lee Wei Lim

**Affiliations:** 1DrLim Neuromodulation Lab, Department of Biosciences and Bioinformatics, School of Science, Xi’an Jiaotong-Liverpool University, Suzhou 215123, China; ypop@connect.hku.hk (P.O.P.Y.); tsuikc@connect.hku.hk (K.C.T.); laquili@apu.ac.jp (L.A.); 2School of Biomedical Sciences, Li Ka Shing Faculty of Medicine, The University of Hong Kong, Hong Kong 999077, China; fungml@hku.hk; 3School and Hospital of Stomatology, Peking University, Beijing 100081, China; hengboonchin@bjmu.edu.cn; 4Faculty of Medicine, Atlas University, Istanbul 34408, Turkey; ersoykocabicak@gmail.com; 5Department of Neurosurgery, Maastricht University, 6229 HX Maastricht, The Netherlands; a.jahanshahi@maastrichtuniversity.nl (A.J.); y.temel@maastrichtuniversity.nl (Y.T.); 6Department of Neuropsychology and Psychopharmacology, Maastricht University, 6200 MD Maastricht, The Netherlands; a.blokland@maastrichtuniversity.nl; 7School of Management, Ritsumeikan Asia Pacific University, Beppu 874-0011, Japan; 8Suzhou Key Laboratory of Neurobiology and Cell Signaling, Department of Biosciences and Bioinformatics, School of Science, Xi’an Jiaotong-Liverpool University, Suzhou 215123, China; avkalueff@gmail.com; 9Department of Anatomy, Faculty of Medicine, University Malaya, Kuala Lumpur 50603, Malaysia; wkahhui@um.edu.my

**Keywords:** DNA methylation, transcutaneous auricular vagus nerve stimulation, memory, cognitive impairment, neuronal activity

## Abstract

Transcutaneous auricular vagus nerve stimulation (taVNS), an emerging noninvasive neuromodulation technique, has shown promise for improving memory. A better understanding of the epigenetic mechanisms underlying the effects of taVNS would inform the molecular outcomes essential for memory and cognition. In this review, we synthesize the current literature on the neurophysiological and biochemical basis of taVNS. Next, we explore how DNA methylation regulators (e.g., DNA methyltransferase 3a) and readers (e.g., methyl-CpG binding protein 2) differentially regulate memory, and how their activity and expression can be regulated by neuronal activity. Finally, we describe the potential involvement of DNA methylation in mediating the memory regulatory effects of taVNS and discuss possible directions for future studies.

## 1. Introduction

Memory can be defined as a storage of our past experiences, which shapes our identity and personality. Information from our experiences is continuously encoded and consolidated into our memories. The initial memories can be fleeting but can be retrieved and re-consolidated into more stable, long-lasting memories, leading to a dynamic and complex integration of our experiences [1,2]. The information is first processed in the short-term store in the prefrontal cortex, which is then transferred and consolidated into long-term memory in the hippocampus [3,4]. Particular memories are stored in specific networks in the brain known as engrams [4,5]. Although there is broad agreement that memories/engrams are stored in the cortex, there are several theories on the processes involved, including the standard consolidation theory, the multiple trace theory, and the scene construction theory [6].

Our ability to encode new memories or recall past episodic memories declines with age, linked to structural and functional changes in memory-related brain regions as we age [5]. Remarkably, the risk of developing dementia is 5.0% among those aged 71 to 79, and increases exponentially to 24.2% in those aged 80 to 89 and to 34.7% in those aged 90 or older. With an increasingly aging population, a better understanding of the underlying mechanisms of age-related episodic memory decline is needed, which could lead to better interventions to prevent or treat memory loss due to old age. Although there has been much research on neuroprotective agents and on the development of pharmaceutical drugs [7,8,9], most of these only treat the symptoms or are limited by inconsistent efficacy and adverse effects [9]. An attractive alternative to these conventional treatments is noninvasive neuromodulation, in which specific brain regions are stimulated to modulate neuronal activity in a safe and personalized manner [10,11].

Different forms of noninvasive neurostimulation have been shown to improve memory in older people [12], including the direct stimulation of specific memory-related brain regions through transcranial magnetic stimulation (TMS), direct current stimulation (DCS), and alternating current stimulation (ACS), or the systematic stimulation of multiple brain regions through transcutaneous auricular vagus nerve stimulation (taVNS). Although these stimulation techniques have demonstrated cognitive-enhancing potential in aging, taVNS is attractive because it is easier to administer and requires less expensive equipment compared to TMS [13]. In taVNS, an electric current with fixed intensity (0.5–50 mA), frequency (between 0.5 and 30 Hz), and pulse width (50–500 μs) is applied at the auricular branch of the vagus nerve (ABVN) in the auricle (e.g., cymba concha, concha, and tragus) [14]. The activated neuronal signals from the ABVN are projected toward the brainstem nuclei and diffusively propagate to memory-related brain regions such as the hippocampus [15,16,17]. Studies demonstrated that taVNS was able to enhance cognitive performance in rodent models [18,19] and humans [13,20] with cognitive impairments. However, the underlying molecular mechanism of taVNS has not been fully investigated. A previous study using an invasive approach to stimulate the vagus nerve (VN) showed that the epigenetic landscape in the brain is altered, which suggests that epigenetic modulation could be an important mediator of the effects of taVNS [21].

During memory encoding, diverse epigenetic changes result in the differential expression of genes that promote or suppress neuronal plasticity, leading to remodeling of the synaptic network and de novo neurogenesis in the hippocampus. The epigenetic landscape in our brain changes as we age, altering our ability to form new neurons and synaptic connections [22,23,24,25]. This is characterized by a global decline in DNA methylation (e.g., 5-mC, 5-hmC) across multiple brain regions, as captured by bisulfite sequencing, immunohistochemistry, chromatin-immunoprecipitation sequencing (CHIP-seq), and next-generation sequencing studies [24,25,26]. The DNA methylation pattern and readouts are regulated by DNA methyltransferases, methyl-binding proteins, and demethylases [24,25,27,28,29]. Although depletion of one of these DNA methylation-related proteins may result in cognitive impairment in humans and animal models, the exact role of each class of protein is not fully understood [30,31,32]. Meanwhile, previous studies demonstrated that altered neuronal activity also modulates DNA methylation and demethylation in genes promoting or suppressing memory formation, facilitating long-term potentiation (LTP) for memory consolidation [33,34,35]. The modulation of neuronal activity in specific brain regions by taVNS was reported to alter the spatiotemporal pattern of DNA methylation, affecting memory performance in rodents [21,24,25].

In this review, we synthesize the findings on how taVNS enhances memory performance in humans and rodents, and how DNA methylation plays an important role in regulating memory. Furthermore, we explore the potential role of DNA methylation in mediating the memory enhancement effects of taVNS.

## 2. Memory Enhancement Effects of taVNS

### 2.1. Anatomical and Physiological Basis of taVNS

The VN, also called cranial nerve (CN) X, is the longest cranial nerve with the most extensive distribution. It is a major component of the parasympathetic nervous system, regulating autonomic, cardiovascular, endocrine, gastrointestinal, immune, and respiratory systems [36,37]. The VN travels from the medulla oblongata in the brainstem, through the jugular foramen, to the left colic flexure, innervating visceral organs (e.g., heart, lungs, and digestive tract) en route [38]. The VN is a mixed nerve (80% afferent fibers and 20% efferent fibers) containing A, B, and C fibers. The A fibers include large myelinated somatic afferent and efferent nerves and small myelinated nerves that transmit afferent visceral information; the B fibers constitute the cranial parasympathetic outflow; and the C fibers are small unmyelinated nerves that transmit afferent visceral information. The cholinergic efferent fibers originate from the dorsal motor nucleus of the VN (DMN) and the nucleus ambiguus (NA), whereas the glutamatergic afferent fibers terminate in the nucleus of the solitary tract (NST, receiving 95% of the afferent projections), the spinal nucleus of the trigeminal nerve (SNT), and the area postrema (AP) [39,40]. After converging into a single trunk, the VN fibers pass through ganglia and diverge into different branches as they travel through the neck to the colon. The VN branches into the auricular, meningeal, sympathetic, pharyngeal, and laryngeal branches in the neck [41], and then continues to innervate all thoracic viscera and structures along the gastrointestinal tract.

The auricular branch of the VN innervates the posterior structures of the outer ear, including the cymba concha, inner wall of the tragus, and concha. The NST is the primary structure in the brainstem that receives afferent signals from the ABVN through the superior ganglion of the VN [17,36]. Following NST activation, afferent signals are directly or indirectly projected to the forebrain, limbic, and brainstem sites, including the SNT, parabrachial area, dorsal raphe, periaqueductal gray, thalamus, amygdala, hippocampus, and neocortex [15,16,17]. Glutamatergic afferent signals from the NST are first relayed to the locus coeruleus (LC, through the nucleus paragigantocellularis), raphe nuclei (RN), pedunculopontine nucleus (PPN), and basal forebrain (BF). Downstream noradrenergic (from LC), serotonergic (from RN), and cholinergic (from PPN and BF) signals are then projected to structures related to memory formation, including the hippocampus and amygdala [42,43,44,45,46,47,48]. The hippocampus, which includes the cornu ammonis 1/2 (CA1/2), CA3, and dentate gyrus (DG, containing the neurogenic subgranular zone), is crucial for long-term, spatial, and recognition memories; hence, modulating its activity may be a way to alter memory performance.

The VN can be directly stimulated via electrodes implanted in the left cervical VN, which activate the VN centers in the brainstem to elicit physiological effects. Given the easy access to cutaneous innervations in the external acoustic meatus, the ABVN is an attractive alternative stimulation target. Unlike direct VN stimulation (VNS), taVNS is a noninvasive method for stimulating the cutaneous receptive field of the ABVN in the outer ear. This leads to afferent signals projected to brainstem nuclei, which then propagate to memory-related structures, including the hippocampus and prefrontal cortex, to modulate neuronal activity (see Figure 1). Notably, the right and left VN innervate visceral organs asymmetrically, with the right VN having a greater influence on heart rate than the left VN [49,50,51,52]; hence, most commercially available taVNS devices target the left cymba concha or concha [53,54].

### 2.2. Therapeutic Effects in Humans and Animals

All VN stimulation approaches share a similar anatomical basis and have demonstrated promising enhancing effects on memory and cognition in humans and animals. Implantable VNS was originally approved by the U.S. Food and Drug Administration (FDA) as a surgical technique for treating drug-resistant epilepsy, depression, and morbid obesity [55]. Its use in treating Alzheimer’s disease (AD) and other age-related cognitive conditions has recently been demonstrated in humans and rodents. Alternatively, noninvasive taVNS has demonstrated potential utility in modulating memory and cognitive performance in healthy individuals, in addition to clinical subjects [20,56]. Despite much research on the therapeutic efficacy of taVNS in humans, there have been limited animal studies on its biochemical mechanisms.

### 2.3. Selective Memory Modulation by taVNS

Various controlled studies and clinical trials have demonstrated that taVNS can improve memory in healthy young and old individuals and patients with cognitive impairment (e.g., mild cognitive impairment, MCI) (see Table 1). Several studies have shown that taVNS administered to the left cymba conchae can enhance recognition memory [57,58,59] and spatial working memory [20] in young individuals. A study by Zhao et al. showed that taVNS intervention enhanced working memory in young individuals with 24 h sleep deprivation [60]. A recent study by Shin et al. showed that taVNS administered during a 2-back task improved non-auditory working memory performance in older adults with age-related hearing loss, which is also a risk factor for dementia [61]. In young individuals, administering taVNS to the left tragus significantly enhanced verbal working memory [62,63], short-term memory [64], and item-order memory [65]. In older individuals, administering taVNS to the left inner tragus enhanced associative memory, as demonstrated by more correct hits in a face–name association task and episodic memory task [13]. In clinical dementia, a 24-week taVNS intervention (stimulation site: left auricular conchae, pulse: 0.3 ms square wave at 20 Hz, duration: 30 s for each pulse, repeated every 5 min) significantly enhanced verbal memory performance in MCI patients [66].

Several clinical studies have demonstrated that taVNS can enhance diverse types of memory in individuals of different ages and with different conditions. Notably, the timing of the taVNS intervention can affect the outcomes and types of memory modulated. Enhancement of different types of memory suggests that taVNS has differential modulatory effects on diverse brain areas, such as the prefrontal cortex involved in working memory storage, the temporal gyri involved in the representation of items in long-term memory during short-term memory processing, and the hippocampus involved in spatial representation and long-term memory encoding. The interactions of these brain regions downstream of taVNS will be discussed in the next section.

In addition to human studies, memory-enhancing effects of taVNS have been demonstrated in young and cognitively impaired animal models. A study by Vázquez-Oliver et al. [67] evaluated taVNS in naive young adult male CD-1 mice (healthy young controls) and Fmr1KO mice (a fragile X syndrome model characterized by learning disabilities and cognitive impairment) in a novel-object recognition test (NORT). After 48 h of the familiarization phase, both animal models treated with taVNS demonstrated superior performance compared to their sham controls, suggesting an improvement in long-term recognition memory. A study by Yu et al. [19] investigated the effects of taVNS in 6-month-old APP/PS1 mice (an Alzheimer’s model) in the Morris water maze (MWM) and NORT. Interestingly, the taVNS group demonstrated enhanced performance in MWM, but no changes in NORT, probably due to the short object interaction time of 20 s during the familiarization phase. The effects of taVNS have also been evaluated in a model of vascular cognitive impairment, which showed the taVNS group had significantly better performance in NORT and Y-maze than the sham controls [18]. Together, these studies suggest that taVNS can enhance long-term memory and spatial memory in rodent models.

### 2.4. Electrophysiological and Biochemical Effects of taVNS

Despite extensive studies on the behavioral outcomes of taVNS in humans and rodents, there have been limited studies exploring the biochemical mechanism of taVNS. At the network level, taVNS is suggested to activate diffuse noradrenergic, serotonergic, and cholinergic systems in the brainstem or subcortical nuclei, thereby modulating glutamatergic and GABAergic circuits in the cortex, limbic structures, and other brain regions (see Figure 1) [44,68,69,70,71,72]. A functional magnetic resonance imaging (fMRI) study revealed that taVNS strengthened the functional connectivity between the LC and hippocampus, altering the upstream temporo-parietal semantic network [73,74]. Typically, connectivity of the left hippocampus to prefrontal regions and the cingulate was increased, while connectivity to the left anterior temporal lobe was decreased [75]. This altered connectivity may affect the conversion of working memory into long-term memory.

These functional enhancements could also be a result of the downstream effects of upregulated neuroplasticity triggered by activated neurotransmitter systems [76]. At the molecular level, upregulation of neurotrophic and growth factors, such as brain-derived neurotrophic factor (BDNF), could result in neuroplasticity effects, including de novo neurogenesis and long-term potentiation for synaptogenesis. To date, studies investigating how taVNS affects neuroplasticity have only been performed in rodent models of stroke, in which taVNS was found to upregulate neuroplasticity markers such as cyclic adenosine monophosphate (cAMP), protein kinase A (PKA), phosphorylated cAMP-response element binding (p-CREB), phospho-endothelial nitric oxide synthase (p-eNOS), BDNF, vascular endothelial growth factor (VEGF), growth differentiation factor 11 (GDF11), α7nAch receptor, GAP-43, and peroxisome proliferator- activated receptor-γ (PPAR-γ) [77]. Stimulating the VN through implanted electrodes has also been shown to mediate neuroplastic effects. For example, acute stimulation of the VN in rats upregulated the expression of Bdnf and fibroblast growth factor (FGF) in the hippocampus and cortex [78,79]. Morphologically, acute VNS triggered increased numbers of BrdU^+^ proliferating cells in the dentate gyrus of the hippocampus [80,81], whereas chronic VNS (1 month) resulted in the persistent enhancement of hippocampal neurogenesis, as marked by increased numbers of BrdU^+^ proliferating cells and Dcx^+^ immature neurons in the dentate gyrus [80]. A study by Sanders et al. further demonstrated that taVNS had memory-enhancing effects associated with a modified epigenetic landscape in the hippocampus and cortical regions [21]. The DNA methylation pattern of memory- and plasticity-related genes such as Bdnf, activity-regulated cytoskeleton (Arc), and DNA methyltransferase 3a (DNMT3a) was differentially altered in these brain regions [23,82]. This highlights the importance of dissecting DNA methylation components for a better understanding of the molecular mechanism of taVNS.

## 3. DNA Methylation and Memory: A Functional Perspective

Memory and DNA methylation are tightly linked, whereby changes in the genomic methylation pattern of different neurons determine the resultant memory networks [83,84]. In mammals, methylation occurs at cytosine nucleotides, in which 5-methyl cytosine (5-mC) is converted to 5-hydroxymethylcytosine (5-hmC). Although the majority of methylated DNA reside in CpG islands (a cytosine nucleotide followed by a guanine nucleotide), methylation in non-CpG sites (e.g., CAC) is also important for regulating memory [85]. DNA methylation involves DNA methyltransferases (DNMTs) and demethylases that regulate methylation homeostasis, and methyl-CpG-binding protein 2 (MeCP2) that acts as the reader of the spatiotemporal methylation pattern [75] (see Table 2). As we age, the brain’s DNA methylation pattern undergoes dynamic changes; this is linked to altered neuroplasticity, leading to compromised memory and learning. Overall, there is a decline in DNA methylation, with CpG islands becoming hypermethylated and intergenic regions becoming hypomethylated, leading to the impaired expression of neurogenic genes (e.g., Bdnf) and increased risk of the expression of memory-suppressing or dementia-related genes [82,86,87]. Meanwhile, in vivo and in vitro evidence shows that neuronal activity also affects the expression of DNA methylation-related proteins, implicating their possible involvement in the effects of taVNS.

### 3.1. DNA Methylation Regulators and Readers

#### 3.1.1. DNA Methyltransferases

Three DNA methyltransferases (DNMT1, DNMT3a, and DNMT3b) have been identified in vertebrates. DNMT1 is involved in maintaining methylation in hemi-methylated DNA, whereas DNMT3a and DNMT3b are important for de novo DNA methylation (i.e., C → 5-mC). In the postnatal stage, DNMT1, DNMT3a, and DNMT3b are highly expressed in the brain. In adults, the level of DNMT3b drops drastically, while DNMT3a and DNMT1 are differentially expressed in different brain regions [114,115], with DNMT3a and DNMT1 highly expressed in the hippocampus (CA1–3 and DG) and DNMT1 highly expressed in the ventromedial hypothalamus and habenula.

Various studies have highlighted the importance of DNMT1 and DNMT3a in regulating memory and cognition. In the human brain, downregulation of nuclear DNMT1 is linked to Parkinson’s disease (PD) and dementia with Lewy bodies (DLB) [88], whereas a mutation in the DNMT3a gene is linked to autism spectrum disorder (ASD) [89,90] and Tatton-Brown–Rahman syndrome (TBRS) [91]. A functional study by Feng, et al. [101] investigated the consequences of single and double knockout of DNMT1 and/or DNMT3a in adult mice. Interestingly, only the double knockout mice had downregulated neuroplasticity genes, attenuated LTP, and a smaller hippocampal size, which was associated with impaired spatial learning and contextual fear memory. Although this study suggests overlapping functionality of DNMT1 and DNMT3a, other studies have shown contrasting results. A study by Morris, et al. [102] showed that conditional knockout of DNMT3a, but not DNMT1, in the forebrain of 6- to 12-week-old male mice resulted in impaired contextual fear memory and spatial–object recognition memory. The DNMT1 knockout mice had only mild LTP changes, whereas the DNMT3a knockout mice showed significantly impaired LTP parameters. Another study showed that DNMT3a and DNMT3b were both upregulated in the dentate gyrus after object-in-place learning, and the use of a DNMT inhibitor impaired long-term object-in-space memory [116].

Indeed, these heterogeneous findings may be due to the complex isoforms of DNMT3a. In mammals, alternative splicing can give rise to two transcript variants of DNMT3a: the predominant full-length DNMT3a1 and DNMT3a2 with truncated N-terminal 219 amino-acid residues [117]. Notably, aged mice showed a decline in hippocampal levels of DNMT1 and DNMT3a2, which was associated with impaired hippocampal-dependent memories [30,103]. Overexpression of DNMT3a2 in the dorsal hippocampus of aged mice compromised object-location long-term memory and increased contextual fear memory [30]. Knockdown of DNMT3a2 in the dorsal hippocampus of 3-month-old mice downregulated synaptic activity-induced expression of Arc and Bdnf, leading to impaired long-term spatial memory and contextual fear memory. Meanwhile, knockdown of Dnmt3a1 in the dorsal hippocampus of 8-week-old mice impaired long-term spatial object recognition and contextual fear memory, whereas overexpression of its downstream mediator neuropilin-1 (Nrp1) restored the behavioral deficits [103]. This process was shown to be associated with neuroplasticity changes, as DNMT3a1 knockdown downregulated various activity-dependent neuroplasticity genes. In vitro studies have shown that DNMT3a targets neuronal enhancers and inter-promotor regions during hippocampal neural stem cell differentiation, upregulating neurogenic genes [118,119].

Additional evidence has also demonstrated that DNMT3a has roles in memory consolidation. The hippocampal DNMT3a mRNA level was found to be enhanced 30 min after training in contextual fear conditioning. Meanwhile, long-term fear memory was impaired by DNMT inhibitors Zebularine and RG108 administered in the hippocampus before and immediately after training [34,120]. Collectively, these findings suggest that DNMT3a is important for regulating hippocampal-dependent memories.

#### 3.1.2. Demethylases

In addition to actively establishing and maintaining methylation sites, demethylation is crucial for regulating the expression of memory-related genes. The demethylation process is catalyzed by the ten-eleven translocation (Tet) family enzymes, Tet1, Tet2, and Tet3. Mutations in the Tet enzymes have been linked to Alzheimer’s disease (AD) and intellectual disability [92,93,94]. All three members of Tet (especially Tet1 and Tet3) are highly expressed in neurons in memory-related brain regions, where they catalyze the conversion of 5-mC to 5-hmC [121]. Although 5-hmC can be converted back to the unmodified cytosine, it is relatively stable and can be recognized by methyl-binding domain-containing proteins such as MeCP2, leading to the transcription of neuronal genes [122]. Interestingly, oxidation leads to MeCP2-binding sites with different affinities, with oxidation at 5-mCpG (5-mCpG → 5-hmCpG) reducing MeCP2 binding, but oxidation at 5-mCpH (5-mCpH → 5-hmCpH) having no effect on MeCP2 binding [123]. Overall, this leads to the upregulation of genes suppressed by MeCP2 binding in CpG islands. Remarkably, Tet enzymes are important in regulating hippocampal neurogenesis in embryonic and adult brain, with Tet1 regulating neural progenitor cell proliferation and synaptic plasticity [104], Tet2 maintaining the pool of adult neural stem cells (NSCs) in aged brain [124], and Tet3 preventing premature differentiation of NSCs into astrocytes [125]. Depletion of any of them disrupts neuroplasticity in the hippocampus and impairs spatial learning and memory [92,104,105,106].

Members of the Gadd45 gene family are also important in mediating the demethylation process [126]. They either convert 5-mC to thymine or convert Tet-mediated 5-hmC to 5-formylcytosine (5fC) and 5-carboxylcytosine (5-caC), followed by thymine excision and base-excision repair [127]. Their relevance to memory consolidation is highlighted by the upregulation of hippocampal Gadd45b mRNA expression after contextual fear conditioning [107]. Interestingly, two Gadd45b knockout studies showed contradictory results, with one study observing impaired long-term contextual fear conditioning [107] and the other showing enhanced contextual fear and spatial memory [108]. A recent study using a mouse model mimicking age-related Gadd45γ decline showed there was impaired hippocampal-dependent memory associated with downregulated early- and late-response genes such as ATF-2, c-Jun, and CREB [32].

#### 3.1.3. Methyl-CpG-Binding Protein 2

Methyl-CpG-binding protein 2, a downstream reader of 5-mC or 5-hmC, is essential for regulating memory in humans and animals. In mammals, MeCP2 is encoded by the *Mecp2/MECP2* gene, whereas its isoforms MeCP2 exon 1 (MeCP2E1) and MeCP2 exon 2 (MeCP2E2) are regulated by alternative splicing, polyadenylation, and posttranslational modifications [128,129]. Following its binding to methylated sites (e.g., mCG, mCAC, hmCG, hmCAC), MeCP2 either recruits chromatin remodelers (e.g., histone deacetylases, histone methyltransferases) or transcription factors (e.g., SOX2, CREB1), which leads to the repression or activation of neuronal genes in a context-specific manner [130,131,132,133]. During development, the protein expression level of MeCP2 continuously increases until 5 weeks in mice and 10 years in humans, when its expression level in neurons becomes comparable to that of histone (~16 × 10^6^ molecules per nucleus) [134,135]. In the adult mouse brain, MeCP2E1 is uniformly expressed in different brain regions, whereas MeCP2E2 is differentially enriched [136].

The importance of MeCP2 in memory regulation is highlighted by its downregulation in dementia conditions and Rett syndrome and its overexpression in MeCP2 duplication syndrome, leading to impaired memory. In humans, loss-of-function mutation of MeCP2 is highly implicated in Rett Syndrome. Individuals with this neurodevelopmental disorder can have impairments in various aspects of memory and cognition, including recognition memory [95], sensory memory [96], selective attention [97], and skill acquisition [98]. Mice with MeCP2 depletion or mutation were found to have worse hippocampal-dependent memory (e.g., spatial memory, contextual fear memory, and social memory) related to impaired electrophysiology and excitatory neuroplasticity in the hippocampus [109,110,111,112]. On the other hand, overexpression of MeCP2 impaired learning and memory [100] in association with deficits in short-term synaptic plasticity and LTP [113]. Studies on AD and dementia suggest a region-specific regulatory role of MeCP2 on memory. A study by Huang et al. showed that MeCP2 mRNA expression was downregulated in the hippocampus of AD patients [31], whereas a study by Lee et al. showed that MeCP2 protein levels were upregulated in the forebrain (putamen and cortex) of AD patients [99]. However, studies in dementia mouse models have shown contrasting results. In senescence-accelerated mouse prone 8 (SAMP8), MeCP2 expression was downregulated in the hippocampus, while overexpressing hippocampal MeCP2 rescued the impaired spatial learning and memory [31]. In APP/PS1 mice, MeCP2 was upregulated in the striatum, contributing to impaired social memory and spatial memory [99], while functional restoration in the hippocampus was associated with enhanced LTP and upregulation of synaptic plasticity genes (e.g., *BDNF, PSD95, NT3*, and *NT4/5*) via CREB1.

Several in vitro and in vivo molecular studies have indicated the involvement of MeCP2 in regulating neurogenesis and neuroplasticity. In adult mouse hippocampus, phosphorylation of MeCP2 at serine 421 modulated the proliferation and differentiation of neural progenitors through the Notch signaling pathway [137]. Meanwhile, DG granule neurons isolated from MeCP2-deficient mice demonstrated delayed neuronal maturation and impaired expression of synaptic genes such as synaptophysin [138]. Impaired neuronal maturation may affect connectivity between the hippocampus and cortical regions, which is important for memory encoding [139]. Furthermore, in neural precursor cells (NPCs), MeCP2 was found to suppress astrocytic differentiation and promote neuronal differentiation, which may be partly mediated via the upregulation of the IκB kinase α-Bdnf pathway [140,141]. Notably, the regulation of MeCP2 on Bdnf and other synaptic genes does not solely rely on mCG sites, while mCH sites (where H = A, C, or T) were found to be potent mediators for epigenetic control [82,142].

### 3.2. Regulation of DNA Methylation by Neuronal Activity

The expression of memory-related genes is affected by the expression and activity of DNMT3a, demethylases, and MeCP2, which are all tightly regulated by neuronal activity (see Figure 2).

#### 3.2.1. Dynamic Modification of the Spatiotemporal Methylation Pattern

Neuronal activity in specific brain regions can be activated by stimulus presentation or by electric stimulation. For memory consolidation, hippocampal LTP, which promotes synaptogenesis and de novo neurogenesis, is an important modulation target of neuronal activity for functional and structural plasticity [143]. The methylation pattern in different neuronal genes can be differentially affected, which shifts the balance between DNMT3a and demethylase expression and activity. Previously, DNMT3a2 was demonstrated to be activity-regulated, where its transcriptional expression is controlled by nuclear calcium signaling [30].

Studies in different species also suggest that glutamatergic activation of N-methyl-D-aspartate receptor (NMDAR) modulates DNMT3a abundance in memory regulation [144,145]. Indeed, the activity and abundance of DNMT3a1 protein in neurons is tightly regulated by synaptic signals, where the activation of GluN2A-containing NMDAR by glutamatergic signals promotes degradation of DNMT3a1 through neddylation via the attachment of small ubiquitin-like peptide neural precursor cell-expressed developmentally downregulated gene 8 (NEDD8) to DNMT3a1 [146]. The reduction in nuclear DNMT3a1 is associated with reduced methylation at the 4th promoter of the *Bdnf* gene, which upregulates BDNF in the hippocampus. Meanwhile, baseline DNMT3a expression in the cortex or hippocampus can be induced by modulating neuronal activity at different brain regions, such as the lateral hemisphere by electroconvulsive therapy (ECT) or the VN by VNS [21,22,23,147]. This leads to differential demethylation (e.g., period homolog 2, CREB binding protein, glutamate receptor interacting protein 1) or de novo methylation (e.g., zinc finger homeobox 2, coiled-coil domain- containing 33) on the promoter or enhancer regions of transcription factors and neuronal genes, with effects persisting beyond 24 h [33].

Similar to DNMT3a, demethylases such as Gadd45b and Tets are also dynamically regulated by neuronal activity [33,121,148]. Neuromodulation studies have shown that Gadd45b can be induced through ECT [149] or deep brain stimulation (DBS) [150]. Methylated DNA immunoprecipitation (MeDIP) analysis revealed that the *Bdnf IX* promoter and *Fgf-1B* regulatory regions could be selectively demethylated, allowing upregulation of neurogenesis-promoting growth factors [149]. In addition to Gadd45b, Tets are also important in mediating neuronal activity-induced DNA methylation changes [33]. Although neuronal activity can boost Tet1 mRNA in the hippocampus and Tet3 mRNA in the cortex, these two demethylases can also reciprocally affect synaptic activity [75,93,121,148,151]. Overexpressing Tet1 or Tet3 upregulated various neuronal activity-regulated genes (e.g., Npas4, c-Fos, Egr2, and Egr4) in the hippocampus, but reduced the expression level of neuronal surface glutamatergic receptor 1 (GluR1) [151,152]. Meanwhile, prelimbic DBS reduced the transcript level of Tet1 and increased the transcript level of Tet3, accompanied by increased expression of *Bdnf IV* and postsynaptic genes (e.g., postsynaptic density protein 95 and CREB) [22]. Overall, DNMT3a and demethylases are differentially regulated by neuronal activity in isoform- and region-specific manners, resulting in active methylation or demethylation in synaptic plasticity- or neurogenesis-related genes.

#### 3.2.2. Differential Binding by MeCP2

In addition to the methylation pattern, the activity of the methylation reader MeCP2 is also important in mediating neuronal activity-related memory modulation. Activity-induced calcium influx through L-type voltage-gated calcium channels (L-VGCCs) and/or NMDAR leads to phosphorylation at MeCP2 serine 421 (S421) and S424 and dephosphorylation at S80 [153,154]. Phosphorylation of MeCP2 S421, mediated via calcium/calmodulin n-dependent protein kinase II (CaMKII) or aurora kinase B, reduces the binding of MeCP2 on *Bdnf* promoter III and enhances its transcription [137,155]. Another study showed that MeCP2 S421 phosphorylation is also important for the proper establishment of inhibitory synapses and neuronal dendrites in the cortex [156]. Apart from *Bdnf*, the binding of MeCP2 to the promoter of *reelin,* a neuronal migration gene important for synaptic plasticity, suppressed its transcription, whereas neuronal activity reciprocally increased *reelin* mRNA [157,158]. On the other hand, MeCP2 S80 dephosphorylation attenuates MeCP2 binding to specific gene promoters, promoting the expression of neuroplasticity genes, including *Rab3d*, *Vamp3*, *Pomc*, *Gtl2,* and *Igsf4b* [153]. In addition to MeCP2 S421 and S424, neuronal activity can also induce phosphorylation at S86, S274, and T308 [159]. The interaction between the MeCP2 repressor domain and the nuclear receptor co-repressor (NCoR) complex is disrupted by T308 phosphorylation, relieving the suppressive action of MeCP2 on activity-regulated genes via NCoR. Notably, studies have mainly investigated neuronal activation in alleviating MeCP2-mediated transcriptional suppression of neuroplasticity genes, whereas studies have barely explored the effects of differentially phosphorylated MeCP2 on activating gene expression [130]. Adding to the complexity, neuronal activity can also induce other post-translational modifications in MeCP2. A study showed that SUMOylation at Lys-412 residue was promoted by neuronal activation and phosphorylation at MeCP2 S421 and T308 [160]. As a downstream effect, the interaction between MeCP2 and CREB is relieved, releasing CREB for transcriptional activation of *Bdnf*. Collectively, neuronal activity alters DNA methylation regulators at the transcriptional and translational levels to control the abundance of methylated or hydroxy-methylated sites, whereas post-translational modifications on MeCP2 differentially regulate its inhibitory action on transcription.

## 4. Perspectives for Future Mechanistic Studies on taVNS

In addition to the direct effect of taVNS on neurotransmitter modulation, its transcriptional effects are also important in mediating and sustaining its therapeutic action. Upon stimulation of the cutaneous ABVN on the outer ear, the electric signal is relayed to brainstem nuclei through afferent fibers of the VN. This signal can then propagate through the diffuse system to other memory-related brain regions, such as the hippocampus and prefrontal cortex. The altered neuronal activity can then trigger dynamic remodeling of the DNA methylation pattern in post-mitotic neurons and neuronal progenitor cells, allowing upregulation of memory-enhancing genes and associated neuroplasticity changes (see Figure 2). A study demonstrated that invasive VNS could enhance recognition memory in rodents, which was associated with an altered epigenetic landscape (upregulated DNMT3a and neuroplasticity genes in the hippocampus) [21]. However, no study has demonstrated a causal relationship between taVNS and altered DNA methylation in the hippocampus. Considering that inhibiting hippocampal DNMT abolished the memory-enhancing effects of prelimbic cortical deep brain stimulation (PrL DBS), depletion of hippocampal DNMT at the mRNA or protein level could abolish taVNS-mediated memory enhancements at the molecular and behavioral levels [22]. Further studies comparing the therapeutic outcome and underlying epigenetic changes of taVNS with other pharmacological interventions could better inform the therapeutic mechanism and efficacy.

The study by Yu et al. showed that taVNS elicited molecular effects at the pathological end of cognitive aging (i.e., Alzheimer’s disease) [19]. However, aging-related cognitive decline is heterogeneous and exists as a spectrum of severity; hence, it would be prudent to evaluate the neuroprotective potential of taVNS across the spectrum of cognitive decline. The complicated process of aging may be attributed to heterogeneous functional alterations in DNA methylation in different brain regions or the pathological processes involved. Although existing studies have demonstrated perturbed expression of DNMT3a, demethylases, and MeCP2 in aged conditions, the exact molecular mechanism involving these DNA methylation regulators and readers is largely unknown. In classical perspectives, upregulation of DNMTs facilitates DNA methylation, which in turn reduces the transcriptional expression of target genes. With the discovery of different isoforms and post-translational modification mechanisms of DNMT3a and MeCP2, their roles in regulating DNA methylation writing and readout and on the memory-modulatory effects will need to be revised. Understanding how MeCP2 differentially activates or suppresses different genes and how different isoforms of DNMT3a regulate memory upon neuronal activation may reveal the epigenetic mechanisms in aging-related memory decline, which would facilitate the development of neuroprotective strategies. Additionally, the expression patterns of proteins and genomic methylation patterns are unique in different brain regions, implying the importance of studying DNA methylation in a brain region-specific manner [114,115,136,161]. Multi-level studies on regional DNA methylation changes triggered by taVNS, including the regulation of writers, erasers, and readers, the differential changes in the genomic methylation pattern, and the transcriptional and translational outcomes, will be crucial to better characterize the complexity of DNA methylation processes in memory regulation. Ultimately, this would facilitate the optimization of the timing and treatment parameters of taVNS at the clinical level.

## 5. Summary

This review highlights the important role of DNA methylation in mediating and sustaining the memory-enhancing effects of taVNS. Further investigations into the effects of taVNS on the various DNA methylation-related proteins (e.g., DNA methyltransferase, demethylase, and methyl-binding proteins) will facilitate our understanding of the underlying neuromodulatory mechanisms of taVNS and the memory regulatory roles of DNA methylation-related proteins. This would advance our knowledge on how changes in DNA methylation impact cognition in the aging context and could translate to the optimization of taVNS stimulation parameters in clinical practice.

## Figures and Tables

**Figure 1 cells-14-01327-f001:**
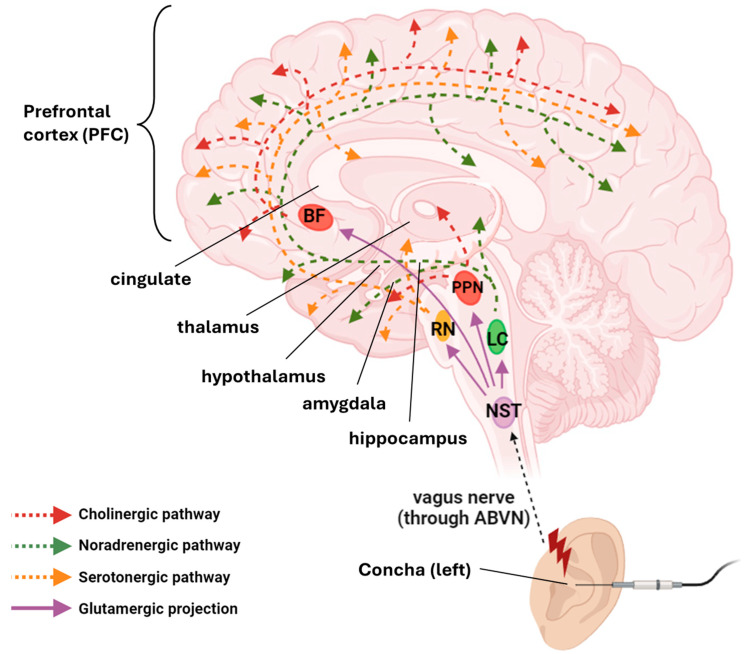
Proposed anatomical pathways activated by taVNS for memory regulation. In taVNS, the cutaneous receptive field of the auricular branch of the vagus nerve (ABVN) is electrically stimulated at the left concha or cymba concha. The signal first projects to the nucleus of the solitary tract (NST) and then to the locus coeruleus (LC), raphe nuclei (RN), pedunculopontine nucleus (PPN), and basal forebrain (BF) nuclei. This results in the activation of noradrenergic, serotonergic, and cholinergic pathways, leading to the modulation of neuronal activity in multiple memory-related brain regions, including the hippocampus, prefrontal cortex (PFC), cingulate cortex, thalamus, hypothalamus, and amygdala. Electrophysiological studies suggest that the functional connectivity between the hippocampus and PFC and cingulate cortex can be strengthened, implicating a more effective conversion of short-term memory to long-term memory. The figure was created with Biorender.com.

**Figure 2 cells-14-01327-f002:**
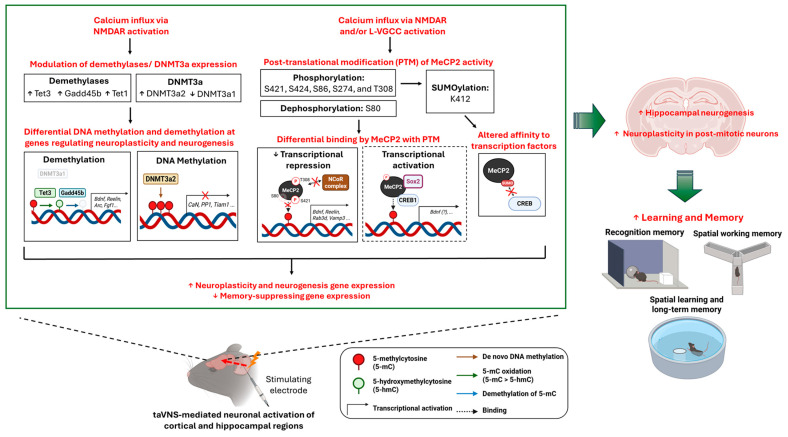
Proposed molecular mechanism underlying the memory regulatory effects of taVNS. Neuronal activity in the hippocampus can be modulated by taVNS, resulting in calcium influx through NMDAR or L-VGCC. This leads to upregulation of DNMT3a2, Tet1, Tet3, and Gadd45b, but downregulation of DNMT3a1 in the nucleus. Upregulation of DNMT3a2 promotes de novo methylation of cytosine residues on ‘memory-suppressing genes’ such as *CaN, PP1,* and *Tiam1*. Downregulation of DNMT3a1, Tet enzymes (converts 5-methylcytosine to 5-hydroxylmethylcytosine), and Gadd45b (removing methylation tag on cytosine) facilitates demethylation on neuroplasticity genes, including *Bdnf, Reelin, Arc,* and *Fgf1*. Neuronal activation also induces post-translational modification of MeCP2, including phosphorylation on multiple serine (S421, S424, S86, S274) or threonine residues (T308), dephosphorylation on S80, and SUMOlyation on lysine 412 (K412). This may (1) alleviate the transcriptional repressive binding of MeCP2 to 5-methylcytosine or 5-hydroxylmethylcytosine, facilitating the transcription of neuroplasticity genes; (2) alter MeCP2 interaction with nuclear co-repressor (NCoR) complex and other chromatin remodelers; (3) weaken the interaction with CREB, increasing its availability for transcriptional activation; or (4) induce activity in the transcriptional activator domain of MeCP2, facilitating the recruitment of transcription factors (e.g., Sox2, CREB1). Together, differential changes in DNA methylation and changes in MeCP2 activity lead to the enhanced expression of neuroplasticity genes. This promotes hippocampal neurogenesis and enhances post-mitotic synaptic plasticity, creating a brain niche and neurocirculatory system that allows more effective learning and memory. Arc, activity-regulated cytoskeleton; BDNF, brain-derived neurotrophic factor; CREB, cAMP-response element binding protein; DNMT3a1, DNA methyltransferase 3a transcript 1; DNMT3a2, DNA methyltransferase 3a transcript 2; Gadd45b, growth arrest and DNA-damage inducible 45b; MeCP2, methyl CpG-binding protein 2; NMDAR, NMDA receptor; L-VGCC, L-type voltage-gated calcium channel; SOX2, SRY-Box Transcription Factor 2; Tet, Ten eleven translocation enzyme. The figure was created with BioRender.com.

**Table 1 cells-14-01327-t001:** Summary of the different types of memory modulated by taVNS in human and animal studies.

Study	Memory Task	taVNS Delivery	Stimulation Site	Subjects	Subject Condition	Effect of taVNS
Human studies						
[13]	Associative memory	During/after learning (online)	Inner side of the tragus	50M/50F (Mean age: 60.57 ± 2.54)	Older adults	Higher correct hits in face-name task (*p* < 0.05)
[57]	Recognition memory	After learning	Left cymba conchae	Young: 49M/51F (Mean age: 22.20 ± 1.97) Old: 29M/70F (Mean age: 55.13 ± 6.59)	Healthy adults	No effect on recall or recognition (*p* > 0.05)
[58]	Recognition memory	During learning (online)	Left cymba conchae	14M/46F (Mean age: 23.39 ± 4.67)	Healthy adults	Increased hits for words remembered with higher subjective confidence (*p* < 0.05)
[65]	Item order memory	During memory task	Left tragus	Control: 10M/23F (Mean age: 19.8) Sham first: 6M/9F (Mean age: 20.4) taVNS first: 4M/10F (Mean age: 20.4)	Healthy adults	Higher accuracy on the order memory task in taVNS groups (*p* < 0.01)
[20]	Spatial working memory	Before/during testing (online/offline)	Left cymba conchae	36M/24F (Mean age: 19.90 ± 1.49)	Healthy adults	Offline taVNS significantly increased hits in spatial 3-back task (*p* < 0.05)
[66]	Auditory verbal learning test-HuaShan version (AVLT-H)	24-week intervention	Auricular acupoints: heart (concha, CO_15_) and kidney (CO_10_)	taVNS: 5M/20F (Mean age: 66.9 ± 3.66) sham: 4M/23F (Mean age: 67 ± 4.36)	Patients with mild cognitive impairment	Significantly increased N5 (immediate recall) and N7 (delayed recall) for taVNS group (*p* < 0.001) after intervention
[62]	Verbal short-term working memory	During learning	Left posterior tragus	taVNS: 3M/9F (Mean age: 19.17 ± 0.88) sham: 4M/8F (Mean age: 20.50 ± 2.48)	Healthy adults	Improved performance in memory questions (*p* = 0.007)
[60]	Working memory	Before testing	Left cymba conchae	19M/16F (Mean age: 21.26 ± 1.90)	Healthy adults with 24-hour sleep deprivation	Improved accuracy rate in spatial 3-back tasks (*p* < 0.05)
[63]	Rey Auditory Verbal Learning Test (RAVLT)	2-week intervention	Left tragus	30M/46F (Mean age: 48.32)	Healthy adults	Significantly improved immediate recall and short-term memory score in taVNS group after intervention (*p* < 0.05)
[64]	Short-term memory	Before testing	Left tragus	9M/9F (Mean age: 73.5 ± 4.71)	Older adults	Significantly reduced total error in 7, 8, 9-digit span tasks after 5-min taVNS (*p* < 0.001)
[59]	Recognition memory	During learning	Left cymba conchae	13M/49F/3NS (Mean age: 24.24 ± 5.25)	Healthy adults	Improved recollection-based memory performance in taVNS group compared to sham 1 week after learning (*p* = 0.001)
[61]	Working memory	During memory task	Cymba conchae (left/right counterbalance)	Typical hearing: 23M/13F (Mean age: 64.03 ± 3.72) Hearing impaired: 14M/6F (Mean age: 65.55 ± 3.66)	Older adults with age-related hearing loss	Improved memory performance in 2-back test for subjects with hearing impairment (*p* < 0.05)
Animal studies						
[67]	Novel object-recognition memory (NORT) for long-term memory	After familiarization phase	Left concha	Naive young-adult male CD-1 mice (10–12 weeks old)		Improved object-recognition memory performance at 48 h (*p* = 0.01)
				Fmr1KO young-adult male CD-1 mice (fragile X syndrome model)		Improved object-recognition memory performance at 48 h (*p* = 0.0003)
[18]	NORT, Y-maze test for spatial memory	Before behavioral tests	Left cymba concha	Vascular Cognitive Impairment model: 8-week-old male C57BL/6 mice with common carotid arteries (CCA) dissected		Higher discrimination index scores in the NOR test (*p* < 0.001) and rates of spontaneous alternations in the Y-maze test (*p* = 0.008)
[19]	Spatial learning and memory (MWM), NORT	Before and during days of behavioral tests	Bilateral auricular concha	6-month-old APP/PS1 mice (Alzheimer’s model)		Longer target quadrant time and more platform crossing in MWM (*p* < 0.05); No change in recognition memory
				6-month-old C57BL/6 mice		No change in long-term spatial memory and recognition memory

Note: M = male, F = female, NS = not specified, NORT = novel object recognition test, MWM = Morris water maze.

**Table 2 cells-14-01327-t002:** Neurocognitive implications of DNA methylation (DNAm)- related proteins in human and animal studies.

DNAm-Related Protein	Function	Disease/Condition	Neurocognitive Implications	Ref
Human studies				
DNMT1	Methylation maintenance	Parkinson’s disease (PD)	Decreased nuclear expression in postmortem human brain samples from PD and DLB patients	[88]
		Dementia with Lewy Bodies (DLB)		
DNMT3a	De novo methylation (converting C to 5-mC)	Autism spectrum disorder (ASD)	Heterozygous mutation of DNMT3A gene is linked to ASD	[89,90]
		Tatton-Brown–Rahman syndrome (TBRS)	Heterozygous DNMT3A variant identified in TBRS patients	[91]
Tet1	Converting 5-mC to 5-hmC	Early-onset Alzheimer’s disease (EOAD)	Loss-of-function mutation of TET1 gene is linked to EOAD	[92]
Tet2		Early-onset Alzheimer’s disease (EOAD)	Loss-of-function TET2 variant is linked to EOAD and FTD	[93]
		Frontotemporal dementia (FTD)		
Tet3		Intellectual disability (ID)	Haploinsufficiency of TET3 due to heterozygous mutation links to ID	[94]
MeCP2	5-mC and 5-hmC binder and reader (can also bind 5-fC and 5-acC with lower affinity)	Rett syndrome	Loss-of-function mutation in MeCP2 links to Rett syndrome, characterized by impaired recognition memory, sensory memory, selective attention, and skill acquisition	[95,96,97,98]
		Alzheimer’s disease (AD)	Downregulated MeCP2 mRNA expression in AD patient hippocampus	[31]
			Upregulated MeCP2 protein expression in AD patient forebrain (putamen, cortex)	[99]
		MeCP2 duplication syndrome	Gain-of-function in MECP2 gene copy causes intellectual disability, developmental delays, and speech difficulties	[100]
Animal studies				
DNMT1 and/or DNMT3a	DNMT1: methylation maintenance DNMT3a: De novo methylation (C → 5-mC)	Mice with double knockout of DNMT1 and DNMT3a in neurons	Only the double knockout mice have impaired spatial learning and contextual fear memory	[101]
		Mice with single knockout of DNMT1 or DNMT3a in neurons		
		Mice with single knockout of DNMT1 or DNMT3a in neurons	Only DNMT3a knockout mice have impaired contextual fear memory and spatial–object recognition memory	[102]
		Aged mice with/without hippocampal DNMT3a2 overexpression	Age-related decline of DNMT1 and DNMT3a2 mRNA expression in hippocampus and cortex is linked to impaired object-location long-term memory and contextual fear memory. Overexpressing DNMT3a2 in hippocampus rescued age-related memory decline	[30]
		Young mice with hippocampal DNMT3a2 knockdown	Depletion of DNMT3a2 in hippocampus impaired object-location long-term memory and contextual fear memory	
		Young mice with hippocampal DNMT3a1 knockdown	Depletion of DNMT3a2 in hippocampus impaired object-location long-term memory and contextual fear memory The memory deficit can be rescued by overexpressing its downstream mediator, neuropilin-1 (Nrp1)	[103]
Tet1	Converting 5-mC to 5-hmC	Young mice with Tet1 mutation	Impairment in spatial learning and short-term memory Altered expression and methylation of genes involved in neural progenitor proliferation, neuroprotection, and mitochondria function, e.g., Galanin, Ng2, Ngb, Kctd14, and Atp5h	[104]
		5xFAD mice (AD model) with/without Tet1 heterozygous mutation	Slightly worse contextual fear memory in mice with Tet1 mutation	[92]
Tet2		Adult mature mice	Age-associated hippocampal Tet2 mRNA decline with loss in differentially 5-hydroxymethylated regions (DhMRs) for neurogenic genes. Overexpressing Tet2 in hippocampus improved associative fear memory acquisition	[105]
		Young mice with hippocampal Tet2 knockdown	Impaired spatial learning and long-term contextual fear memory	
Tet3		Young mice with conditional knockout of Tet3 in neuronal cells	Impaired spatial learning	[106]
Gadd45b	Convert 5-mC to thymine Convert 5-hmC to 5-formylcytosine (5fC) and 5-carboxylcytosine (5-caC)	Gadd45b knockout mice	Impaired long-term contextual fear conditioning	[107]
			Enhanced contextual fear and long-term spatial memory, with better long-term potentiation in hippocampus	[108]
Gadd45γ		Aged mice	Reduced Gadd45γ mRNA level in dorsal hippocampus (while not Gadd45α and Gadd45β)	[32]
		Young mice with hippocampal Gadd45γ knockdown	Impaired object–space memory and contextual fear memory	
MeCP2	5-mC and 5-hmC binder and reader (can also bind 5-fC and 5-acC with lower affinity)	Mice with *Mecp2* loss-of-function mutation (Rett syndrome model)	Impaired spatial memory, contextual fear memory, and social memory related to impaired electrophysiology and excitatory neuroplasticity in the hippocampus	[109,110,111,112]
		Mice with MeCP2 overexpression in neurons (MeCP2 duplication syndrome model)	Impaired contextual fear memory and novel object recognition, associated with deficits in short-term synaptic plasticity and LTP	[113]
		Senescence-accelerated mouse prone 8 (SAMP8) (AD model)	Reduced hippocampal MeCP2 expression (mRNA and protein level) Overexpressing hippocampal MeCP2 rescued deficits in spatial learning and retention memory	[31]
		Amyloid precursor protein (APP)/presenilin1 (PS1) transgenic mice (AD model)	Increased striatal MeCP2 expression (mRNA and protein level). Knockdown of striatal MeCP2 rescued deficits in social memory and spatial memory in 10-month-old mice	[99]

## Data Availability

No new data were created or analyzed in this study. Data sharing is not applicable to this article.

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
