# Peer review of "DNA Methylation: A Potential Mediator of the Memory Regulatory Effects of taVNS"

_cells, 2025, doi:10.3390/cells14171327_

Round 1
Reviewer 1 Report
Comments and Suggestions for Authors
This review manuscript focus on taVNS, an emerging noninvasive neuromodulation technique, which has shown promising effects on improving memory. This work summarizes recent research and review literatures on taVNS, bridging the therapeutic efficacy observed in rodent models and human patients, neurophysiological and biochemical basis, and molecular mechanisms underlying positive memory effects with a special focus on DNA methylation and its key regulators. Please address below comments to better convey the main message to the audience who are not familiar with taVNS and DNA methylation:
- In line 54-55, is there any recent research/review article on neuroprotective agents and pharmaceutical drugs developed in the past 20+ years to prevent or treat memory loss, in addition to the Lockhart & Lestage, 2003 reference? A more recent, comprehensive review on the conventional pharmacotherapy might help highlight the importance of developing alternative interventions such as noninvasive neuromodulation.
- In the Introduction section, it might be helpful to add in the discussion on why taVNS represents an attractive noninvasive brain stimulation method compared to the other approaches outlined in the Goldthorpe et al., 2020 reference (e.g. PFC as stimulation site)? Is it because of the unique downstream projection of taVNS neuronal network? Better therapeutic effects? Or more convenient intervention in the out ear for electrical stimulation? Please elaborate on why taVNS was chosen as the main topic for this review manuscript.
- In section 2.3, line 172, Table 1 is missing from the main text. It may help better serve the article to summarize the therapeutic effects by taVNS observed in both human (clinical trials) and animal models (research studies).
- In section 2.3, line 181-184. It seems that among all taVNS studies on human subjects, Wang et al., 2022 study is the only one that reports disease patients (MCI) at the clinical dementia level, whereas other studies focus on either healthy aged adults or healthy young adults. Is that correct? If so, the therapeutic effects of memory enhancement by taVNS observed in healthy population might not be translated to the disease states, considering the heterogeneity of different pathology, severity, cognitive decline in different neurological conditions, such as Alzheimer's, frontal lobe dementia, Parkinson's, or ischemic stroke.
- In section 3, it may help better serve the article to summarize current knowledge on DNA methylation regulators affected by taVNS observed in both human (clinical trials) and animal models (research studies), in table.
- The authors argue that DNA methylation factors decline/change as we age, how would those factors change, including but not limited to mRNA level, protein level, PTM, in different neurological conditions? Therefore, targeting those DNA methylation factors through taVNS may be beneficial to improve memory in both healthy and disease populations?
- In section 4, it may be helpful to discuss the neuroprotective effects of taVNS, which alters DNA methylation landscape and reprograms neuronal activity gene, compared to the conventional pharmacotherapies such as small molecule inhibitors or gene therapy as monotherapy, if there is any study reported.
The English language is fine and generally does not require any improvement.
Author Response
Comments 1: [In line 54-55, is there any recent research/review article on neuroprotective agents and pharmaceutical drugs developed in the past 20+ years to prevent or treat memory loss, in addition to the Lockhart & Lestage, 2003 reference? A more recent, comprehensive review on the conventional pharmacotherapy might help highlight the importance of developing alternative interventions such as noninvasive neuromodulation.]
Response 1: [We sincerely appreciate the reviewer’s insightful feedback on citing more updated review article for highlighting the importance of non-invasive neuromodulation. In response, we have added two review articles (Huang et al., 2023, Zhang et al., 2024) that summarized the currently used medications and ongoing clinical trials for Alzheimer’s Diseases (AD), and highlighted the limitation of pharmacological approach (i.e. only for symptom management, inconsistent efficacy, adverse effects) in our text. Please refer to line 52-55 in our revised manuscript.
Reference:
Zhang, J.; Zhang, Y.; Wang, J.; Xia, Y.; Zhang, J.; Chen, L. Recent advances in Alzheimer’s disease: mechanisms, clinical trials and new drug development strategies. Signal Transduction and Targeted Therapy 2024, 9, 211, doi:10.1038/s41392-024-01911-3.
Huang, L.-K.; Kuan, Y.-C.; Lin, H.-W.; Hu, C.-J. Clinical trials of new drugs for Alzheimer disease: a 2020–2023 update. Journal of Biomedical Science 2023, 30, 83, doi:10.1186/s12929-023-00976-6.]
Comments 2: [In the Introduction section, it might be helpful to add in the discussion on why taVNS represents an attractive noninvasive brain stimulation method compared to the other approaches outlined in the Goldthorpe et al., 2020 reference (e.g. PFC as stimulation site)? Is it because of the unique downstream projection of taVNS neuronal network? Better therapeutic effects? Or more convenient intervention in the out ear for electrical stimulation? Please elaborate on why taVNS was chosen as the main topic for this review manuscript.]
Response 2: [We sincerely appreciate the reviewer’s constructive suggestion for elaborating the choice of taVNS as the review topic. In response, we have added two sentences to highlight the systematic modulatory effect by taVNS and its easier administration compared to other techniques. Please refer to line 58-65 in our revised manuscript.]
Comments 3: [In section 2.3, line 172, Table 1 is missing from the main text. It may help better serve the article to summarize the therapeutic effects by taVNS observed in both human (clinical trials) and animal models (research studies).]
Response 3: [We would like to thank the reviewer for pointing out the absence of Table 1. We noted Table 1 may be accidentally deleted in the previous version for peer review. In response, we have added Table 1 back in the revised manuscript.]
Comments 4: [In section 2.3, line 181-184. It seems that among all taVNS studies on human subjects, Wang et al., 2022 study is the only one that reports disease patients (MCI) at the clinical dementia level, whereas other studies focus on either healthy aged adults or healthy young adults. Is that correct? If so, the therapeutic effects of memory enhancement by taVNS observed in healthy population might not be translated to the disease states, considering the heterogeneity of different pathology, severity, cognitive decline in different neurological conditions, such as Alzheimer's, frontal lobe dementia, Parkinson's, or ischemic stroke.]
Response 4: [We thank the reviewer for the critical question on the clinical efficacy of taVNS in diseased population. We agree that at clinical dementia level, only Wang et al. (2022)’s study has evaluated the cognitive-modulatory effect of taVNS, and they obtained positive result (improved MOCA-B, N5, and N7 scores), supporting taVNS can be beneficial to the clinical patients. We agree that the neurodegenerative diseases are highly heterogeneous, and the therapeutic effect of taVNS can be complicated by the pathologies and disease stages. Therefore, our review also explores the neuroprotective role of taVNS, focusing on those middle- or old-aged adults who have not developed clinical dementia yet (but slight decline in memory or cognitive functioning may have been started) and see whether taVNS can also improve memory in this group to prevent further degenerative process. Our review has included studies with a broad age range or primarily focused on the old-aged groups (i.e. >60), where in both groups, taVNS treatment can improve memory performance in a design-specific manner. In the revised manuscript, we have added a recent study (Shin et al., 2025) showing taVNS boosts the non-auditory working memory performance in old individuals with age-related hearing loss (the condition is a risk factor for dementia) (line 173-175). This may serve as supporting evidence for the neuroprotective potential of taVNS in vulnerable population. Please refer to Table 1 for an overview of the listed studies.
Reference:
Wang, L.; Zhang, J.; Guo, C.; He, J.; Zhang, S.; Wang, Y.; Zhao, Y.; Li, L.; Wang, J.; Hou, L.; et al. The efficacy and safety of transcutaneous auricular vagus nerve stimulation in patients with mild cognitive impairment: A double blinded randomized clinical trial. Brain Stimulation 2022, 15, 1405-1414, doi:https://doi.org/10.1016/j.brs.2022.09.003.
Shin, J.; Noh, S.; Park, J.; Jun, S.B.; Sung, J.E. Transcutaneous auricular vagus nerve stimulation enhanced working memory in older adults with age-related hearing loss. Scientific Reports 2025, 15, 26629, doi:10.1038/s41598-025-11363-6.]
Comments 5: [In section 3, it may help better serve the article to summarize current knowledge on DNA methylation regulators affected by taVNS observed in both human (clinical trials) and animal models (research studies), in table.]
Response 5: [We sincerely appreciate the reviewer’s constructive suggestion on using a table to summarize our current understanding of the DNA methylation regulators. In response, we have created a table to summarize the human and animal studies regarding the expression and functional roles of the DNA methylation-related proteins in neurocognitive or aging conditions (please see Table 2). Hopefully this will give a clearer insight of why it is important to study the expression pattern and activity of the DNA methylation proteins as downstream targets of taVNS for memory enhancement.]
Comments 6: [The authors argue that DNA methylation factors decline/change as we age, how would those factors change, including but not limited to mRNA level, protein level, PTM, in different neurological conditions? Therefore, targeting those DNA methylation factors through taVNS may be beneficial to improve memory in both healthy and disease populations? ]
Response 6: [We would like to thank the reviewer for the insightful feedback regarding the relevance of the DNA methylation factors on neurological and healthy populations. We have added Table 2 to summarize how the DNA methylation factors may be dysregulated in neurocognitive conditions. Briefly, DNA methyltransferases (e.g. DNMT1, DNMT3a), Tet demethylase family, and MeCP2 (methylation reader) tend to be downregulated in aged brains (both human and mouse). Genetic mutation of either one the factors is also associated with various neurocognitive conditions, including Alzheimer’s disease, Parkinson’s disease, intellectual disability, etc. In mouse model, many of these proteins are demonstrated to be important for long-term memory, and their expression or activity can be modulated by neuronal activity. Since studies in human and animal also showed taVNS treatment can modulate different types of memory including long-term memory, these DNA methylation-related proteins may play a role in mediating the memory enhancement effects by taVNS, especially for long-term memory (see Table 1). Further studies will be needed to validate this point, but if we know the mechanism, we may be able to optimize the population for receiving the treatment and timing of stimulation. We are highlighting this point in line 539-544 of the revised manuscript.]
Comments 7: [In section 4, it may be helpful to discuss the neuroprotective effects of taVNS, which alters DNA methylation landscape and reprograms neuronal activity gene, compared to the conventional pharmacotherapies such as small molecule inhibitors or gene therapy as monotherapy, if there is any study reported.]
Response 7: [We agree with the reviewer’s comment that comparing taVNS with conventional pharmacotherapies would inform its neuroprotective effect and mechanism. However, the neuroprotective potential of taVNS is still under research. Limited clinical studies have evaluated its utilization in clinical and risky population, and no animal study has compared it with existing pharmacotherapies yet. In response, we have added a sentence (line 517-519) in Section 4 to highlight it as a direction for future study.]
Reviewer 2 Report
Comments and Suggestions for Authors
The manuscript submitted to the Cells editorial office with ID (Cells-3778541) by Yee et al. is a fascinating review paper that analyzes the contribution of DNA methylation to the regulation of memory processes, based on the latest research.
In the first chapters, the authors focus on a non-invasive brain stimulation method involving transcutaneous auricular vagus nerve stimulation (taVNS) to improve memory processes. Hence, the effects of taVNS stimulation on anatomical connections to memory-related brain areas, such as the hippocampus, were presented. The behavioral, electrophysiological, and biochemical effects of this stimulation were then analyzed.
Finally, changes in DNA methylation were considered in the context of memory disturbances.
In general, the study was well-designed, and the structure of the individual chapters is clear and comprehensive.
Due to the innovative approach to the problem of memory processes, I recommend this interesting work for publication.
Author Response
Full comment: [
The manuscript submitted to the Cells editorial office with ID (Cells-3778541) by Yee et al. is a fascinating review paper that analyzes the contribution of DNA methylation to the regulation of memory processes, based on the latest research.
In the first chapters, the authors focus on a non-invasive brain stimulation method involving transcutaneous auricular vagus nerve stimulation (taVNS) to improve memory processes. Hence, the effects of taVNS stimulation on anatomical connections to memory-related brain areas, such as the hippocampus, were presented. The behavioral, electrophysiological, and biochemical effects of this stimulation were then analyzed.
Finally, changes in DNA methylation were considered in the context of memory disturbances.
In general, the study was well-designed, and the structure of the individual chapters is clear and comprehensive.
Due to the innovative approach to the problem of memory processes, I recommend this interesting work for publication.]
Response: [We would like to thank the reviewer for the positive comments and recommendation.]
Reviewer 3 Report
Comments and Suggestions for Authors
This review is a type of “narrative review." It explores an important issue regarding the potential role of DNA methylation as a mediator of the memory-regulating effects of transauricular vagus nerve stimulation (taVNS). Overall, this review article is well written. The number of references cited is substantial, and the reference list adequately covers the relevant literature.
I have only minor comments:
- The sentence on page 9 (lines 406-407) “Studies in different …….” needs rephrasing. As it stands now, this is too much of a simplification. It's not NMDAR that modulates DNMT3a, but glutamic acid acting through NMDA receptors. Please modify this sentence accordingly.
- Including a list of abbreviations used in the manuscript would make it much easier for the reader to read the text.
Author Response
Comments 1: [The sentence on page 9 (lines 406-407) “Studies in different …….” needs rephrasing. As it stands now, this is too much of a simplification. It's not NMDAR that modulates DNMT3a, but glutamic acid acting through NMDA receptors. Please modify this sentence accordingly.]
Response 1: [We sincerely appreciate the reviewer’s insightful comment regarding the wordings used to describe DNMT3a expression modulation. In response, we have modified this sentence as “Studies in different species also suggest that glutamatergic activation of N-methyl-D-aspartate receptor (NMDAR) modulates DNMT3a abundance in memory regulation”. Please refer to line 411-412.]
Comments 2: [Including a list of abbreviations used in the manuscript would make it much easier for the reader to read the text.]
Response 2: [We would like to thank for reviewer for the constructive suggestion on improving the readability of our manuscript. In response, we have added a list of abbreviation before reference section of the revise manuscript.]
Reviewer 4 Report
Comments and Suggestions for Authors
This is an interesting review ms. The following comments should be addressed.
- It would be good to include subdivisions in #1 and 2.1.
- There seems little information included with regards to how DNA methylation is analyzed. It would help adding that and explaining the difference between bisulfite sequencing versus using antibodies against 5-mC and 5-hmC.
- It would help to add some more good-looking figures.
- There are various studies showing the association between DNA methylation in brain and environmental challenges that are associated with behavioral performance. These studies should be described.
- The relationships between increases and decreases in DNA methylation and RNAseq illustrate the complexity of these relationships. That should be discussed.
Author Response
Comments 1: [It would be good to include subdivisions in #1 and 2.1.]
Response 1: [We appreciate the reviewer’s suggestion on including subdivisions in sections 1 and 2.1. After careful evaluation, we determined that adding subdivisions in sections 1 (Introduction) and 2.1 (Anatomical and physiological basis of taVNS) may not be the best way to organize the sections. For section 1, we aim to provide an overview of the connections between memory, taVNS, and DNA methylation; adding subdivisions may segregate the 3 subjects-of-interest. For section 2.1, which includes 3 paragraphs in the revised version, we first discussed the basic anatomical and physiological basis of vagus nerve in the 1st paragraph, followed by its relevance to modulate brain activity in the 2nd paragraph, and choice of auricular branch at the outer ear as the stimulation site in the 3rd paragraph. All 3 paragraphs are closely linked and organizing them as a single section should be more comprehensible. We also noted that the font size of the figure legend of Figure 1 is similar to main text and potentially confuse reader as part of the main text. In response, we have amended its positioning (line 144-154).]
Comments 2: [There seems little information included with regards to how DNA methylation is analyzed. It would help adding that and explaining the difference between bisulfite sequencing versus using antibodies against 5-mC and 5-hmC.]
Response 2: [We appreciate the reviewer’s suggestion on elaborating the analyzing method for DNA methylation. We agree including a brief discussion on the analytical techniques may facilitate further discussion on DNA methylation. In response, we have added a sentence in section 1 to briefly describe how DNA methylation is analyzed in various studies to let readers have an overview of the techniques (line 80-83). While further elaborating the difference between those techniques may make the review article more informative, it may not be the most relevant to our aim of providing a theoretical overview of how the DNA methylation regulators (i.e. the mechanism underlying the altered DNA methylation pattern) may mediate the taVNS effects on memory enhancement.]
Comments 3: [It would help to add some more good-looking figures.]
Response 3: [We sincerely appreciate the reviewer’s valuable feedback regarding the number of figures. After careful evaluation, we decided to modify Figure 2 to make it more informative and attractive, and added Table 2 to summarize the neurocognitive implications of each class of DNA methylation-related proteins (drivers for DNA methylation changes). Hopefully this can help convey our idea better.]
Comments 4: [There are various studies showing the association between DNA methylation in brain and environmental challenges that are associated with behavioral performance. These studies should be described.]
Response 4: [We thank the reviewer’s insightful suggestion on including studies demonstrating the relationship between brain DNA methylation and behavior. In response, we have added two sentences in the first paragraph of section 3 (line 258-263) to highlight the change in DNA methylation pattern in brain under neurocognitive and aging conditions. We have also included animal studies highlighting the behavioral changes under the dysfunction of each protein (see Table 2). Hopefully this can let readers appreciate more about the importance of studying DNA methylation in aging brain, thus digging further into our discussions on those proteins that regulate DNA methylation in next subsection.]
Comments 5: [The relationships between increases and decreases in DNA methylation and RNAseq illustrate the complexity of these relationships. That should be discussed.]
Response 5: [We sincerely appreciate the reviewer’s constructive suggestion on highlighting the complicated relationship between DNA methylation and transcriptional outcome. We have modified our sentences in section 4 to highlight the complicated relationship between DNA methylation and RNA expression, and the importance of further study to capture how unique DNA methylation pattern in different regions leads to different transcriptional outcomes. Please refer to line 528-533 and 536-544.]
Round 2
Reviewer 4 Report
Comments and Suggestions for Authors
The authors did a fine job addressing the raised comments. No additional revisions are requested from this reviewer.